# Myocardial Tissue Characterization in Patients with Hypertensive Crisis, Positive Troponin, and Unobstructed Coronary Arteries: A Cardiovascular Magnetic Resonance-Based Study

**DOI:** 10.3390/diagnostics13182943

**Published:** 2023-09-14

**Authors:** Mohammed A. Talle, Anton F. Doubell, Pieter-Paul S. Robbertse, Sa’ad Lahri, Philip G. Herbst

**Affiliations:** 1Division of Cardiology, Department of Medicine, Faculty of Medicine and Health Sciences, Stellenbosch University and Tygerberg Hospital, Cape Town 7505, South Africa; 2Department of Medicine, Faculty of Clinical Sciences, College of Medical Sciences, University of Maiduguri and University of Maiduguri Teaching Hospital, Maiduguri 600004, Nigeria; 3Division of Emergency Medicine, Department of Medicine, Faculty of Medicine and Health Sciences, Stellenbosch University and Tygerberg Hospital, Cape Town 7505, South Africa

**Keywords:** hypertensive crisis, elevated cardiac troponin, unobstructed coronary arteries, myocardial tissue characteristics, cardiovascular magnetic resonance imaging

## Abstract

Hypertensive crisis can present with cardiac troponin elevation and unobstructed coronary arteries. We used cardiac magnetic resonance (CMR) imaging to characterize the myocardial tissue in patients with hypertensive crisis, elevated cardiac troponin, and unobstructed coronary arteries. Patients with hypertensive crisis and elevated cardiac troponin with coronary artery stenosis <50% were enrolled. Patients with troponin-negative hypertensive crisis served as controls. All participants underwent CMR imaging at 1.5 Tesla. Imaging biomarkers and tissue characteristics were compared between the groups. There were 19 patients (63% male) with elevated troponin and 24 (33% male) troponin-negative controls. The troponin-positive group was older (57 ± 11 years vs. 47 ± 14 years, *p* = 0.015). The groups had similar T2-weighted signal intensity ratios and native T1 times. T2 relaxation times were longer in the troponin-positive group, and the difference remained significant after excluding infarct-pattern late gadolinium enhancement (LGE) from the analysis. Extracellular volume (ECV) was higher in the troponin-positive group (25 ± 4 ms vs. 22 ± 3 ms, *p* = 0.008) and correlated strongly with T2 relaxation time (*r_s_* = 0.701, *p* = 0.022). Late gadolinium enhancement was 32% more prevalent in the troponin-positive group (82% vs. 50%, *p* = 0.050), with 29% having infarct-pattern LGE. T2 relaxation time was independently associated with troponin positivity (OR 2.1, *p* = 0.043), and both T2 relaxation time and ECV predicted troponin positivity (C-statistics: 0.71, *p* = 0.009; and 0.77, *p* = 0.006). Left ventricular end-diastolic and left atrial volumes were the strongest predictors of troponin positivity (C-statistics: 0.80, *p* = 0.001; and 0.82, *p* < 0.001). The increased T2 relaxation time and ECV and their significant correlation in the troponin-positive group suggest myocardial injury with oedema, while the non-ischaemic LGE could be due to myocardial fibrosis or acute necrosis. These CMR imaging biomarkers provide important clinical indices for risk stratification and prognostication in patients with hypertensive crisis.

## 1. Introduction

Hypertensive crisis remains a common reason for emergency room visits and is associated with increased major adverse cardiovascular events (MACE) [1,2]. An important risk factor for increased MACE in this population is increased cardiac troponin levels. The reported prevalence of raised cardiac troponin in patients with hypertensive crisis is variable, mainly because cardiac troponin is not assessed routinely in the setting of hypertensive crisis [3,4].

Positive cardiac troponin and unobstructed coronary arteries are associated with both acute myocardial infarction (as defined in the fourth universal definition and including both type 1 and type 2 infarcts) and acute myocardial injury (a troponin leak not otherwise fulfilling the definition of myocardial infarction, typically due to an absence of associated evidence of ischaemia), and both can occur in patients with hypertensive crisis with or without left ventricular hypertrophy (LVH) [5]. Acute myocardial injury is partly mediated through LVH, which is prevalent in hypertensive crisis and accounts for the depleted myocardial oxygen reserve and increased risk of myocardial ischaemia associated with hypertensive heart disease (HHD) [6] without the requirement of epicardial coronary obstruction. In addition, the LV filling pressure increase associated with a sudden rise in the augmentation index (an effect based on enhanced and early systolic reflection waves from the aorta back to the LV, resulting in a diastolic pressure increase in hypertensive patients with elevated systemic vascular resistance) following an acute blood pressure rise can cause myocardial injury even without associated LVH [7]. Other factors that can cause a cardiac troponin rise in hypertensive crisis and are included in the concept of cardiac acute hypertension-mediated organ damage (HMOD) include acute heart failure/acute pulmonary oedema and myocardial infarction [4]. Coronary artery disease and altered cardiac structure and function have been reported in patients with hypertensive crisis and in patients with type 2 myocardial infarction [8,9]. One study reported obstructive coronary artery disease in up to two-thirds of patients with hypertensive crisis and elevated cardiac troponin undergoing cardiac catheterization [9].

Cardiovascular magnetic resonance (CMR) is the gold standard for evaluating cardiac structure and function but also has the unique ability of myocardial tissue characterization. Guidelines on the evaluation of myocardial infarction with unobstructive coronary arteries (MINOCA) recommend CMR for determining aetiological diagnosis [10,11]. The most common aetiologies established in MINOCA using CMR are myocarditis and Takotsubo cardiomyopathy [11]. Although systemic hypertension is common in patients with MINOCA [12], hypertensive crisis and acute cardiac HMOD do not feature prominently as aetiologic diagnoses associated with the presentation. This may be related to the relatively low prevalence of hypertensive crisis in general or because hypertensive crisis is considered the clinical cause; therefore, it will not fulfil the criteria for a MINOCA and will not enter the typical MINOCA diagnostic pathway [10]. Notwithstanding this, patients with hypertensive crisis and elevated troponin with unobstructed coronary arteries share features of MINOCA presentation, and significant overlap is possible between the groups. In addition to its role in establishing aetiologic diagnoses, a recent study demonstrated the utility of early CMR imaging in risk stratification and prognostication of patients with a MINOCA presentation [13].

The prognostic implications of raised cardiac troponin in patients with hypertensive crisis have been established in many studies [8,14]. Despite the prognostic significance and the clinical relevance of establishing an aetiologic diagnosis, the myocardial tissue characteristics in hypertensive crisis with elevated troponin and unobstructed coronary arteries are not clearly defined. Therefore, we set out to determine the myocardial tissue characteristics in such a cohort and compared these cases with a troponin-negative hypertensive crisis control cohort. Furthermore, we assessed the role of structural and functional alterations of the heart and of serum biomarkers in predicting whether patients would be troponin-positive with unobstructed coronary arteries.

## 2. Materials and Methods

### 2.1. Study Population

Adult patients aged 18 years and above with hypertensive crisis referred to Tygerberg Hospital in the Western Cape province of South Africa were enrolled in this study. Diagnosis of hypertensive crisis was based on a systolic blood pressure of ≥180 mmHg and/or a diastolic BP of ≥110 mmHg [15]. All participants were evaluated in line with guideline recommendations, including high-sensitive cardiac troponin T (hs cTnT) and N-terminal prohormone of brain natriuretic peptide (NT-proBNP). hs cTnT was measured at baseline in all participants and repeated 6 to 24 h after obtaining the first assay. hs cTnT values above 14 ng/mL, representing the 99th percentile of the upper reference limit of normal, were considered elevated [5].

Acute myocardial infarction was diagnosed as rising and/or falling plasma levels of hs cTcT with at least one value above the 99th percentile of the upper reference limit with features of myocardial ischaemia. Those without features of myocardial ischaemia were considered to have acute myocardial injury [5]. Patients fulfilling the criteria for myocardial infarction but without evidence of atherothrombosis or obstructed coronary arteries (>50% stenosis in at least one major epicardial coronary artery) on angiography were deemed to have had type 2 myocardial infarction [5]. Participants with positive troponin and unobstructed coronary arteries (acute myocardial injury and type 2 myocardial infarction) constituted the study group, while the troponin-negative hypertensive crisis patients served as controls.

The following exclusion criteria were applied: hypertensive disorders of pregnancy, altered level of consciousness, inability or refusal to consent, and patients under 18 years of age. This study was approved by the Health Research Ethics Committee (HREC) of Stellenbosch University, and all participants granted a written consent. The Declaration of Helsinki was adhered to.

### 2.2. Cardiac Magnetic Resonance Imaging

All participants underwent CMR imaging using a 1.5 T scanner (MAGNETOM Avanto, Siemens Healthcare, Erlangen, Germany) within 48 h of presentation to Tygerberg Hospital. The scanning protocol included breath-held, ECG-gated balanced steady-state free precession (bSSFP) cine; T2-weighted short-tau inversion recovery (STIR); parametric mapping (T1 and T2); extracellular volume (ECV) estimation; and late gadolinium enhancement (LGE) imaging [16]. Native and postcontrast T1 mapping were obtained using the appropriate Modified Look-Locker Inversion recovery (MOLLI) sequences [17], while bSSFP was used to acquire T2 mapping. Late gadolinium enhancement images were obtained using an inversion-recovery sequence 10–12 min following intravenous administration of gadolinium-based contrast (0.2 mL/kg of Gadovist (Bayer Pharma AG, Leverkusen, Germany)). Individual inversion time for nulling the myocardium was optimized using an inversion time-scout sequence. Images for this study were acquired in different orientations, including two-, four-, and three-chamber long-axis views and short-axis stack covering the entire left ventricle from base to apex.

### 2.3. Image Analysis

The postprocessing of images was performed by M.A.T. using CVi^42^, a commercially available software (Circle cardiovascular Imaging, version 5.13.10 (2678), Calgary, AB, Canada). Enrolment into this study lasted two years, with an interval of six months from the last participant to the time the imaging data were analysed. The images were anonymised to minimise further reporting bias, and postprocessing was carried out for the group as a single cluster without being classified as cases and controls. Ventricular epicardial and endocardial contours were drawn in end-diastolic and end-systolic phases to determine cardiac end-diastolic volume (EDV), end-systolic volume (ESV), LV ejection fraction, and LV mass [18]. Papillary muscle was included in the LV mass. The biplane area-length method was used to determine the left atrial volume (LAV), and normality or otherwise of cardiac volumes and LV ejection fraction was based on published CMR reference values [18]. Left ventricular hypertrophy was defined using LV mass indexed to body surface area greater than the 95th percentile of CMR reference values for age and gender [18].

To determine the T2-weighted signal intensity (SI), T1 mapping and T2 mapping, endocardial and epicardial contours were drawn in the relevant sequence’s basal-, mid-, and apical short-axis slices. Epicardial and endocardial contours were carefully drawn to avoid contamination from blood pool and epicardial fat, and offsets of 20% and 10% were applied from the epicardial and endocardial borders, respectively. To determine the T2-weighted myocardial: skeletal SI ratio, a region of interest was drawn in the serratus anterior muscle in the same slice with myocardial contouring [19]. To estimate ECV, a blood pool contour was drawn in the basal-, mid-, and apical short-axis slices of corresponding native and postcontrast T1 maps, avoiding the papillary muscle and trabeculations. A blood sample for haematocrit measurement was obtained within 24 h of the CMR study to calculate the ECV using the native and postcontrast T1 mapping [20].

Late gadolinium enhancement was assessed qualitatively in conjunction with other sequences analysing the same cardiac segment and was considered present when visualized in 2 orthogonal planes or in 2 adjacent slices. The pattern of LGE was classified into infarct-pattern LGE (subendocardial or transmural signal consistent with a coronary distribution), or non-ischaemic LGE (focal, midmyocardial, patchy, right ventricular (RV) insertion point) [21,22]. The modified Lake-Louise criteria were applied in suspected cases of myocarditis [23]. Areas with wall motion abnormality, myocardial oedema, and subendocardial/transmural LGE with microvascular obstruction were considered acute myocardial infarction [24]. We have previously published our data on interobserver variability of CMR analysis, demonstrating good to excellent reproducibility of morphological, functional, and tissue characterization parameters [25,26].

### 2.4. Statistical Analysis

Sample size and power calculation leveraged the reproducibility of measurements using CMR imaging for cardiac volumes, mass, and LV ejection fraction [27]. Data were analysed using SPSS version 28 (SPSS Inc., Chicago, IL, USA). The normality of continuous variables was assessed using the Shapiro–Wilk method. Continuous variables are presented as mean ± SD or median (IQR) and compared using independent sample *t*-test, Mann–Whitney test, and 1-way ANOVA with post hoc analysis as appropriate. Categorical variables are presented as proportions and percentages and compared using Fisher’s exact test. The correlation of continuous variables was determined using Spearman’s correlation (two-tailed *p* value), while binary logistic regression analysis was used to determine factors associated with troponin-positive unobstructed coronary arteries. The area under the receiver operator characteristics (ROC) curve (AUC) was used to determine the predictors of troponin-positive unobstructed coronary arteries. GraphPad Prism version 6.04 for MacOS (GraphPad Software 10.0.0 (131), www.graphpad.com, accessed on 12 July 2023) was used to make scatter plots. A predetermined *p* value of <0.05 (two-tailed) was considered significant.

## 3. Results

### 3.1. Clinical and Demographic Characteristics

Eighty-two patients with hypertensive crisis were screened for enrolment into the study (Figure 1). Fifty-eight (71%) had elevated hs cTnT, and twenty-five (43%) of these cases had coronary angiography based on clinical indication. Nineteen (76%) of the patients who underwent coronary angiography had unobstructed coronary arteries and were included in the study. Twenty-four patients with hypertensive crisis and normal cardiac troponin levels served as a control group. The clinical profile and exemplar CMR images of the study participants are presented in Table 1 and Figure 2.

The troponin-positive group with unobstructed coronary arteries were on average 10 years older than the troponin-negative group (57 ± 11 years vs. 47 ± 14 years, *p* = 0.015), and 63% were men. Blood pressure, duration of hypertension, and the rate of nonadherence to antihypertensive medication were similar in the groups. The hs cTnT, serum creatinine, haemoglobin, and NT-proBNP levels were higher in the troponin-positive group. Chronic kidney disease (CKD, *p* = 0.037) and dyslipidaemia (*p* = 0.011) were more prevalent in the troponin-positive group; however, there were more patients with diabetes in the troponin-negative group (*p* = 0.034). Fifteen (79%) of the troponin-positive, unobstructed coronary artery group had a 20% change in cardiac troponin level (delta 20%) on repeat measurement compared to baseline values.

Chest pain (*p* < 0.001), cough (*p* = 0.005), shortness of breath (*p* = 0.009), New York Heart Association classes III and IV, and palpitations (*p* < 0.001) were more prevalent in the troponin-positive group. Angiotensin-converting enzyme inhibitor or receptor blocker (*p* = 0.034), beta-blocker (*p* = 0.002), platelet inhibitor (*p* < 0.001), and statin (*p* < 0.001) were more frequently prescribed in the troponin-positive group. However, there were more patients in the troponin-negative group on calcium channel blockers (*p* = 0.004).

The corrected QT interval tended to be higher in the troponin-positive group (*p* = 0.051), but the prevalence of QTc prolongation was similar. T-wave inversion was more prevalent among the troponin-positive group than the control group (81% vs. 38%, *p* = 0.009). Only one patient with a neurological emergency had T-wave inversion.

### 3.2. Cardiovascular Magnetic Resonance Imaging

Findings from CMR are presented in Table 2 and Appendix A and Figure 2 and Figure 3.

#### 3.2.1. Morphology and Function

The indexed LV end-diastolic and end-systolic volumes were higher in the troponin-positive unobstructed coronary artery group. Similarly, the indexed left atrial volume was 13 mL/m^2^ higher in the troponin-positive group than the troponin-negative group (*p* < 0.001). Using LV ejection fraction, impaired LV systolic function was 36% more prevalent in the troponin-positive group (53% vs. 17%, *p* = 0.012). The prevalence of LVH did not differ in the two groups (*p* = 0.221); however, indexed LV mass was 20 g/m^2^ higher in the troponin-positive group (*p* = 0.030), indicating a larger degree of LVH than the troponin-negative group.

#### 3.2.2. T2-Weighted Signal Intensity Ratio and Parametric Mapping

T2-weighted SI ratios at the basal-, mid-, and apical LV levels were similar (Table 2 and Appendix A). Although the global native T1 measured 16 ms higher in the troponin-positive unobstructed coronaries group, this difference was not statistically significant. However, the troponin-positive unobstructed coronary artery group had a significantly longer T2 relaxation time than the troponin-negative group at the basal-, mid-, and apical LV levels. The maximum difference was noted for global (47 ± 2 ms vs. 49 ± 2 ms, *p* = 0.004) and mid-ventricular (48 ± 2 ms vs. 50 ± 3 ms, *p* = 0.003) T2 relaxation time. The difference in mid-ventricular T2 relaxation time was maintained after excluding patients with infarct-pattern LGE from the troponin-positive group. Global ECV (23 ± 3 ms vs. 25 ± 4 ms, *p* = 0.047) and mid-ventricular (22 ± 3 ms vs. 25 ± 4 ms, *p* = 0.008) ECV were significantly increased in the troponin-positive group when compared to the troponin-negative group. Unlike T2 relaxation time, the difference in ECV was not maintained after excluding cases with infarct-pattern LGE from the analysis. T2 relaxation time correlated significantly (Table 3) with native T1 (*r_s_* = 0.623, *p* = 0.004) and ECV (*r_s_* = 0.645, *p* = 0.044) in the troponin-positive unobstructed coronary artery group. However, there was no significant correlation between native T1 and ECV (*r_s_* = 0.177, *p* = 0.625). None of the imaging biomarkers correlated with hs cTnT (Appendix A).

Appendix A compares the troponin-positive unobstructed coronary artery with a delta 20% and the troponin-negative control group. Although T2 relaxation times and ECV were numerically higher in the delta 20% group compared to the controls, only the apical T2 relaxation time (*p* = 0.044) and mid-ventricular ECV (*p* = 0.020) were statistically significant. Non-ischaemic LGE occurred similarly in the troponin-negative group and the group with delta 20% (*p* = 0.681).

#### 3.2.3. Late Gadolinium Enhancement

A contrasted CMR study was conducted with 41 participants (95%). Two participants from the troponin-positive group were excluded from the contrasted study due to concerns about acute kidney injury. Overall, 63% of the participants displayed LGE. Among the troponin-positive group with unobstructed coronary arteries, infarct-pattern LGE was noted in 29%, while 53% had a non-ischaemic LGE (midmyocardial-3, patchy-1, RV insertion point-4, and subepicardial-1); the remaining 18% had no LGE. Although LGE was 32% more prevalent in the troponin-positive group (82% vs. 50%, *p* = 0.050), the difference in prevalence was not maintained after excluding participants with infarct-pattern LGE (53% vs. 50%, *p* = 0.445). All the troponin-negative cases with LGE had non-ischaemic LGE (midmyocardial-2, RV insertion point-8, patchy-2).

Appendix A compares CMR imaging and serum biomarkers in the group with troponin-positive unobstructed coronary arteries based on the pattern of LGE. Indexed LV end-diastolic (*p* = 0.019) and end-systolic (*p* = 0.029) volumes were higher, while LV ejection fraction was lower in the group with non-ischaemic LGE than the group with ischaemic LGE (*p* = 0.029). Although hs cTnT tended to be higher in the group with infarct-pattern LGE, the difference did not reach statistical significance (*p* = 0.060). Similarly, indexed LV mass and maximum wall thickness were numerically higher in the group with non-ischaemic LGE, but the difference did not attain statistical significance (*p* = 0.083 for both). Native T1 relaxation time, T2 relaxation time, ECV, and T2-weighted SI ratio did not differ in the groups with infarct-pattern vs. non-ischaemic LGE.

### 3.3. Binary Logistic Regression of Factors Associated with Troponin-Positive Unobstructed Coronary Arteries

The results of binary logistic regression for factors associated with troponin-positive unobstructed coronary arteries are presented in Table 4. Of all the factors associated with troponin-positive unobstructed coronary arteries in univariate analysis, indexed LV EDV (OR 1.1, *p* = 0.026) and T2 relaxation time (OR 2.1, *p* = 0.043) maintained an independent association in the multivariable logistic regression model, with a Nagelkerke R square of 0.577. The model correctly classified 82.9% of cases (87.5% of the troponin-negative group and 76.5% of the troponin-positive group).

### 3.4. Receiver Operator Characteristic Curve for Predictors of Troponin-Positive Unobstructed Coronary Artery

The receiver operator characteristic curve and corresponding area under the curve for the imaging biomarkers in predicting troponin-positive unobstructed coronary artery are presented in Figure 4. Indexed LV EDV (AUC 0.80, *p* = 0.001) and indexed LAV (AUC 0.82, *p* < 0.001) were the best CMR imaging predictors of troponin-positive unobstructed coronary artery, followed by mid-ventricular ECV (AUC 0.77, *p* = 0.006), mid-ventricular T2 relaxation time (AUC 0.73, *p* = 0.003), indexed LV mass (AUC 0.71, *p* = 0.018), global T2 relaxation time (AUC 0.71, *p* = 0.009), and LV ejection fraction (AUC 0.70, *p* = 0.028). Native T1 (AUC 0.61, *p* = 0.221) and global ECV (AUC 0.705, *p* = 0.067) did not differentiate the troponin-positive group from troponin-negative controls. The sensitivity, specificity, likelihood ratios, and cut-offs associated with the AUC of the different parameters are presented in Table 5.

## 4. Discussion

Acute myocardial injury and type 2 myocardial infarction occur as HMOD in hypertensive crisis and unobstructed coronary arteries. Increased T2 relaxation time and ECV, and their correlation in the troponin-positive group, could indicate myocardial injury with oedema, while non-ischaemic LGE could be interpreted as either myocardial fibrosis or acute necrosis. We used CMR to assess myocardial tissue characteristics in hypertensive crisis patients with troponin-positive, unobstructed coronary arteries. The main findings are as follows: (1) T2 relaxation time and ECV were significantly increased in the troponin-positive unobstructed coronary artery group. Although native T1 time was numerically higher in the troponin-positive unobstructed coronary artery group, this did not differ statistically. (2) Increasing indexed LV mass and LV EDV, systolic dysfunction, and T2 relaxation times were significantly associated with troponin-positive unobstructed coronary arteries. (3) There was a trend for a higher prevalence of LGE, mainly driven by infarct-pattern LGE that occurred in 29% of the troponin-positive group. (4) No infarct-pattern LGE occurred in the troponin-negative group. Furthermore, indexed LV EDV and LAV demonstrated a diagnostic utility for predicting troponin-positive unobstructed coronary arteries using the ROC curve, followed by ECV and T2 relaxation time.

Although the troponin-positive group was, on average, ten years older than the troponin-negative group, our cohort was younger and with fewer comorbidities compared to previous research, and this may partly explain the observed lower prevalence of obstructive coronary disease [8,28]. Higher serum creatinine at baseline and increased prevalence of CKD among the troponin-positive group reflect the coexistence of impaired renal function and hypertensive crisis [29]. Chronic kidney disease is known to be associated with increased cardiac troponin, NT-proBNP, and T2 relaxation time [30].

The increased T2 relaxation time in the troponin-positive unobstructed coronary artery group is evidence of myocardial oedema. T2 relaxation time is a sensitive indicator of intracellular and extracellular water (acute free water) and provides a reliable, pixel-wise quantitative measure of acute myocardial oedema [20]. Although sensitive to both free water and water bound to macromolecules, the increase in T2 relaxation time associated with free water during acute cellular injury is 40 times higher than bound water [31]. This makes elevated T2 signal well suited to detect tissue characteristics associated with acute myocardial injury. Perivascular fibrosis, microvascular disease, and endothelial dysfunction are prevalent in hypertensive crisis and can cause myocardial ischaemia, increasing free intracellular water. Prolonged myocardial ischaemia can affect the integrity of membrane-bound Na^+^/K^+^-ATPase, leading to increased intracellular Na^+^ concentration and an osmotic gradient that promotes vasogenic interstitial oedema and interstitial expansion [32].

The ECV fraction is a nonspecific measure of interstitial expansion related to various pathologies, including diffuse interstitial fibrosis, infiltrative diseases, and interstitial myocardial oedema [20]. The significant correlation of T2 relaxation time with ECV among our group with troponin-positive unobstructed coronary arteries is highly suggestive of myocardial oedema with or without associated myocardial necrosis as a cause for the increased ECV in that group. Native T1 time is a sensitive marker of increased myocardial water content with a better diagnostic utility for myocardial oedema than T2 imaging and is expected to increase along with T2 relaxation time in the setting of myocardial oedema [20]. However, native T1 did not significantly differ between the troponin-positive group and controls in our study. The similarity of native T1 time may be related to the myocardial fibrosis associated with HHD prevalent in both groups and to the sensitivity of T1 to both acute oedema states and fibrosis-related interstitial expansion. An overlap between interstitial expansion related to chronic myocardial fibrosis and the acute myocardial oedema related to acute HMOD may increase native T1 time, diluting the T1 signal difference between groups, especially in milder increases in myocardial water content. This may also be partly responsible for the lack of correlation observed between native T1 time and ECV among the troponin-positive group.

Although the increase in T2 relaxation time could be due to the oedema associated with myocardial infarction, the troponin-positive unobstructed coronary artery group maintained significantly higher T2 relaxation time and ECV after excluding cases of infarct-pattern LGE. The ischaemic imbalance associated with hypertensive crisis could cause myocardial injury and/or type 2 myocardial infarction, resulting in myocardial edema and an increased T2 relaxation time and ECV. T2 relaxation time showed an independent association with troponin-positive unobstructed coronary arteries (Table 4), differentiating the group from the troponin-negative controls with an AUC of 0.71 to 0.73 (Table 5). Leveraging the established prognostic roles of T2 relaxation time in other cardiac conditions associated with myocardial oedema [13,33], the increased T2 time observed in the group with troponin-positive group indicates an increased risk of MACE; however, this will require a confirmation in a longitudinal outcome study. Interestingly, the level of hs cTnT did not significantly differ in the troponin-positive unobstructed coronary artery group with and without infarct-pattern LGE (Appendix A), which may be related to several reasons. The rise in cardiac troponin may have been relatively modest if the areas of infarction were small. Secondly, the high prevalence of non-ischaemic LGE in the troponin-positive group may be related to areas of focal necrosis, resulting in more substantial cardiac troponin elevation rather than the milder elevation associated with subtle myocardial injury. Thirdly, significant overlap occurs between acute myocardial injury and type 2 myocardial infarction, and subtle cases of myocardial infarction not apparent on LGE (subthreshold) may be classified as myocardial injury but may mask T1-detectable myocardial expansion associated with infarction [5]. Other factors contributing to the lack of a clear distinction in hs cTnT include the sample size, resulting in a type 2 statistical error, and the timing of cardiac troponin assay relative to the onset and duration of myocardial injury and infarction.

Most CMR-based studies involving patients with elevated troponin and unobstructed coronary arteries did not include patients with hypertensive crisis, making comparison with this study difficult [11]. However, increased T2 relaxation time and ECV without an accompanying increase in native T1 time, similar to our findings, was reported in patients with acute cardiac allograft rejection compared to those without [34,35]. Similarly, they demonstrated a significant correlation of ECV with T2 relaxation time but not with native T1. Myocardial oedema is common in cardiac transplantation and is associated with poor prognosis in acute cardiac allograft rejection [33]. However, the prognostic implications of myocardial oedema have not been evaluated in hypertensive crisis.

Late gadolinium enhancement is the gold standard imaging method for demonstrating myocardial replacement fibrosis and is instrumental in establishing aetiologic diagnoses in many cardiac pathologies [36]. The reported prevalence rates for non-ischaemic LGE in hypertensive patients are variable due to the heterogeneity of the population [37,38,39,40]. However, these studies did not include hypertensive crisis. Acute myocardial injury and type 2 myocardial infarction are associated with an increased risk of MACE more than type 1 myocardial infarction [41], and the high prevalence of all forms of LGE observed in this study may contribute to the risks. Although generally considered a marker of myocardial fibrosis, LGE is not exclusively seen in fibrosis and can occur in acute myocardial necrosis [42]. Similarly, the vascular changes associated with a hypertensive crisis may result in areas of acute myocardial necrosis, elevated cardiac troponin, and focal/patchy LGE. However, considering previous studies that reported LGE in patients with hypertension in general, the non-ischaemic LGE observed in our cohort may be unrelated to hypertensive crisis [38,39,43], especially if taken in the context of the lack of a statistically significant difference for LGE, native T1, and the prevalence of LVH between the groups.

Furthermore, increased LV volumes and reduced ejection fraction among the group with non-ischaemic LGE, compared to those with infarct-pattern LGE, point to a chronic cardiac remodelling associated with longstanding uncontrolled hypertension.

Indexed LV EDV and left atrial volume demonstrated diagnostic utility for troponin-positive unobstructed coronary artery. Other significant predictors of being troponin-positive with unobstructed coronary arteries included LV mass, LV ejection fraction, and T2 relaxation time, a sensitive marker of increased myocardial water. As alluded to earlier, an increased augmentation index in a hypertensive crisis can cause a sudden increase in LV filling pressure [7]. Previous studies have demonstrated increased cardiac troponin levels among patients with all forms of heart failure [44,45]. Similarly, using CMR imaging, Bularga et al. reported a high rate of structural and functional cardiac alterations in patients with type 2 myocardial infarction [9]. However, their cohort did not include hypertensive crisis. The troponin-positive group in this study demonstrated lower LV ejection and increased LAV, a good surrogate for LV diastolic dysfunction. The diagnostic utility of the imaging biomarkers for troponin-positive unobstructed coronary arteries, as demonstrated by the AUC, may be related to raised troponin independent of coronary artery status. However, cardiac troponin did not correlate with the imaging biomarkers (Appendix A).

Our findings provide evidence for myocardial interstitial expansion, oedema, and increased prevalence of LGE in patients with hypertensive crisis, raised troponin, and unobstructed coronary arteries. Raised troponin and unobstructed coronary arteries encompass acute myocardial injury and type 2 myocardial infarction, two seemingly separate entities with common prognostic implications. However, differentiating them can be challenging in clinical practice [5]. The pathophysiologic processes in hypertensive crisis could result in either or both, and whether they are indeed separate entities or represent a continuum in the spectrum of acute cardiac HMOD remains unclear. The increased risk of adverse events associated with this presentation may be related to myocardial oedema and LGE. Given the uncertainties surrounding the management of myocardial injury and type 2 myocardial infarction in general, our finding lends credence to subjecting patients with hypertensive crisis and troponin-positive unobstructed coronary arteries to CMR, similar to the current recommendations for the MINOCA population. In their recent publication, Bergamaschi et al. [13] reported %LGE and T2 relaxation time to independently predict MACE in patients with a MINOCA at approximately three years of follow-up. Although more than 50% of their cohort had hypertension, they did not include patients with hypertensive crisis. A follow-up study will be required to determine the prognostic implications of T2 relaxation time and the presence of LGE in our cohort.

Limitations: Our findings should be interpreted in light of the following limitations. It was a single-center study with a limited sample size with unequal distribution of males and females, resulting in a type 2 statistical error, especially in some subgroup analyses. Similarly, the number of covariates included in the regression model relative to the sample size may cause overfitting bias.

Coronary angiography was obtained when clinically indicated and was not conducted in the troponin-negative control group. This limited application of coronary imaging excluded many patients with hypertensive crisis and positive troponin from the study. In addition, intravascular ultrasound, fractional flow reserve, and optical coherence tomography were not included in the study, and therefore, plaque rupture or erosion was not systematically assessed.

Coronary vasospasm could result in myocardial ischaemia and increased troponin. Although a provocative assessment of the coronary arteries was not carried out as part of the study, none of the cohort demonstrated ST-segment elevation at presentation, during the hospital stay, or during coronary angiography. Also, myocardial perfusion imaging, which could have identified myocardial ischaemia and provided evidence of microvascular dysfunction, was not carried out as part of this study.

Follow-up CMR was not carried out to determine the effect of treatment of hypertensive crisis on the structural and functional alterations as well as myocardial oedema, especially since myocardial stunning could complicate hypertensive crisis.

Despite identifying infarct-pattern LGE, a significant overlap could be present in cases of acute myocardial injury and type 2 myocardial infarction, making the differentiation difficult.

Finally, postprocessing of CMR images was conducted unblinded, which could result in some bias. These limitations notwithstanding, the key findings from this study provide insights into myocardial tissue architecture that will be useful in risk stratification and prognostication in patients with hypertensive crisis and troponin-positive unobstructed coronary arteries. Further studies will be required to confirm our findings and determine the prognostic implications of the altered structure, function, and myocardial oedema.

## 5. Conclusions

One-third of the troponin-positive hypertensive crisis group had CMR evidence of myocardial infarction, but this does not fully explain the observed differences in T2 relaxation time and ECV. The increased T2 relaxation time and ECV and their significant correlation in the troponin-positive group suggest myocardial injury with oedema, while the non-ischaemic LGE could be due to myocardial fibrosis or acute necrosis. In addition, the increased T2 relaxation time, ECV, and presence of LGE may contribute to the MACE associated with hypertensive crisis, myocardial injury, and type 2 myocardial infarction. Our study provides evidence for the clinical utility of CMR imaging in risk stratification and prognostication of patients with hypertensive crisis. However, longitudinal research is required to corroborate these findings and determine the prognostic implications.

## Figures and Tables

**Figure 1 diagnostics-13-02943-f001:**
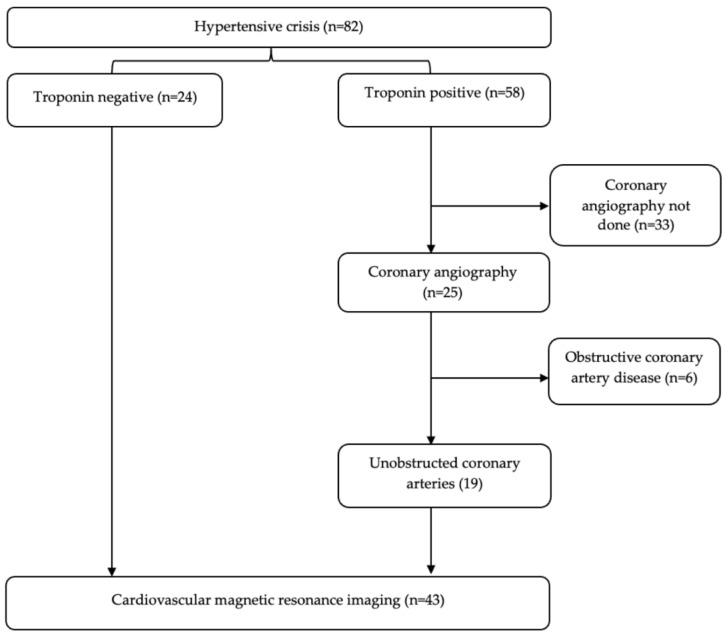
Study flow chart.

**Figure 2 diagnostics-13-02943-f002:**
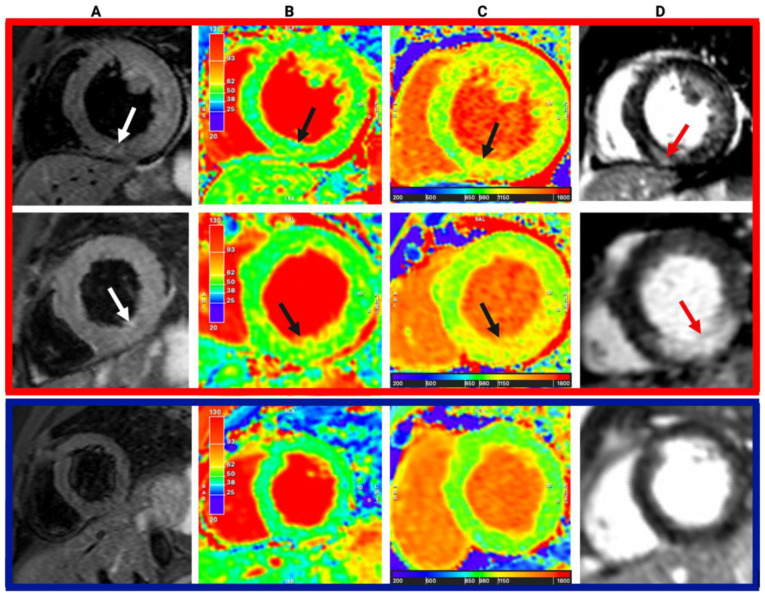
Exemplar CMR images of aetiologic diagnoses. Columns: (**A**) T2-weighted imaging; (**B**) T2 mapping; (**C**) native T1 mapping; (**D**) late gadolinium enhancement. Arrows point to areas of increased signal. Top and middle rows: troponin-positive; bottom row: troponin-negative.

**Figure 3 diagnostics-13-02943-f003:**
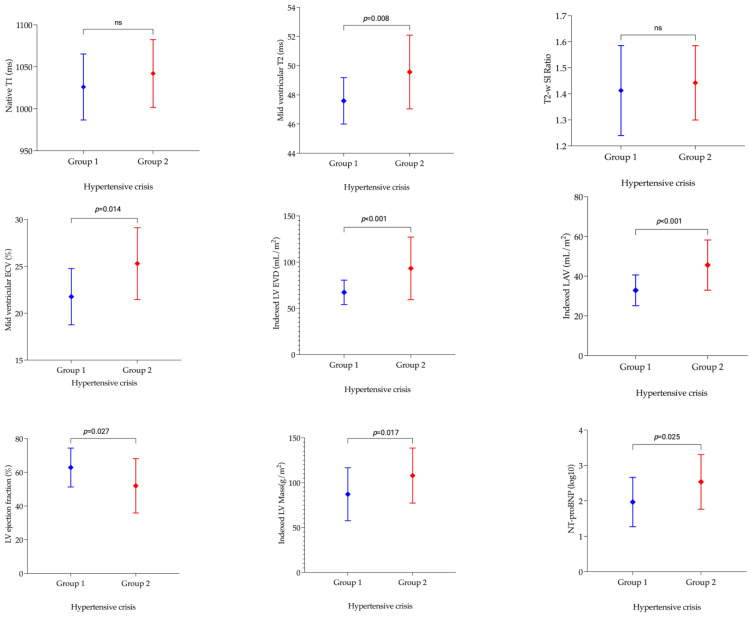
Comparison of imaging and serum biomarkers in the groups with troponin-positive unobstructed coronary arteries and troponin-negative hypertensive crisis. Group 1, troponin-negative hypertensive crisis; Group 2, troponin-positive unobstructed coronary arteries. ECV, extracellular volume; T2-w SI, T2-weighted signal intensity; LV EDV, left ventricular end-diastolic volume (indexed); LAV, left atrial volume (indexed); LV, left ventricular; NT-proBNP, N-terminal prohormone of brain natriuretic peptide; ns, not significant.

**Figure 4 diagnostics-13-02943-f004:**
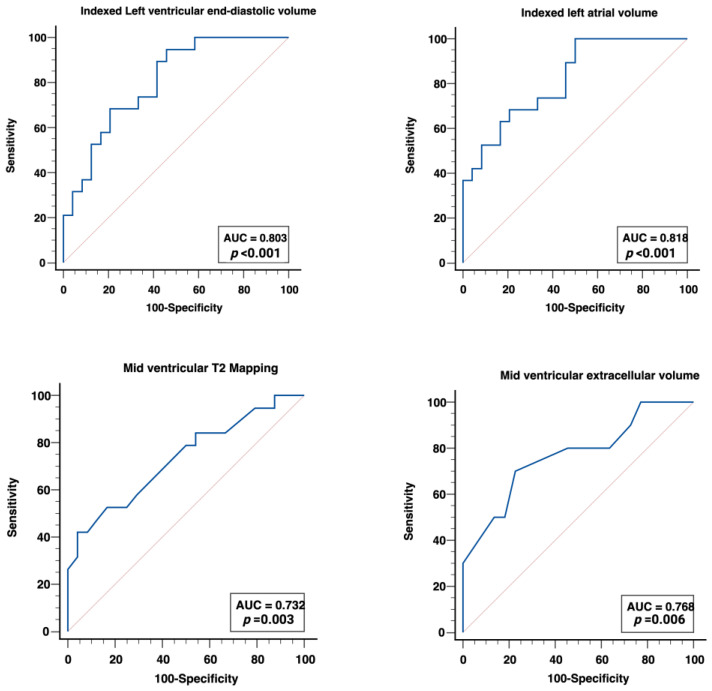
Receiver operator characteristic curve showing the area under the curve (AUC) for various biomarkers in predicting troponin-positive unobstructed coronary artery.

**Table 1 diagnostics-13-02943-t001:** Clinical and demographic characteristics.

Variable	Hypertensive Crisis (*n =* 43)		*p*
	Troponin-Negative Controls (*n =* 24)	Troponin-Positive Unobstructed Coronary Arteries (*n =* 19)	
**Age (years)**	47 ± 14	57 ± 11	0.015
Females, *n* (%)	16 (67)	7 (37)	0.052
Systolic blood pressure (mmHg)	218 ± 28	204 ± 28	0.103
Diastolic blood pressure (mmHg)	125 ± 19	124 ± 21	0.871
Nonadherence, *n* (%)	10/17 (59)	5/15 (33)	ns
Creatinine (µmol/L)	83 (69 to 101)	104 (91 to 117)	0.004
Haemoglobin (g/L)	13.8 ± 1.5	15.1 ± 2	0.002
hs cTnT (ng/L)	10 (6 to 11)	130 (46 to 219)	<0.001
NT-proBNP (ng/L)	77 (22 to 396)	433 (99 to 1448)	0.025
QRS duration	88 (77 to 102)	94 (83 to 106)	ns
QTc	451 (432 to 470)	465 (443 to 515)	0.051
LVH by SL, *n* (%)	6/21	8/16	0.045
LVH by CV, *n* (%)	11/21	6/16 (38)	ns
ST depression, *n* (%)	4/21	6/16 (38)	ns
T-wave inversion, *n* (%)	8/21 (38)	13/16 (81)	0.009
Deep T-wave inversion, *n* (%)	2/21 (10)	6/16 (38)	0.030
Pathological Q wave, *n* (%)	1/21	0/16	ns
*Comorbidities:*			
Duration of hypertension (years)	12 (8 to 26)	12 (3 to 12)	ns
Previous MI, *n* (%)	0 (0)	1 (5)	ns
Chronic kidney disease, *n* (%)	1 (4.2)	5 (26)	0.037
Diabetes mellitus, *n* (%)	5 (21)	0 (0)	0.034
Dyslipidemia, *n* (%)	3 (13)	9 (47)	0.011
Smoking, *n* (%)	6 (25)	5 (26)	ns
*Symptoms and clinical findings:*			
Chest pain, *n* (%)	8 (33)	18 (95)	<0.001
Cough, *n* (%)	2 (8)	9 (47)	0.004
Shortness of breath, *n* (%)	4 (17)	11 (58)	0.005
NYHA class III–IV, *n* (%)	1 (4)	9/19 (47)	<0.001
Palpitations, *n* (%)	7 (29)	16 (84)	<0.001
*Cardiac medications:*			
Aldosterone antagonist, *n* (%)	3 (13)	1 (5)	ns
Beta-blocker, *n* (%)	11 (46)	19 (100)	<0.001
Calcium channel blocker, *n* (%)	21 (88)	9 (47)	0.004
Loop diuretic, *n* (%)	3 (13)	6 (32)	ns
RAS inhibitor, *n* (%)	19 (79)	19 (100)	0.034
Thiazide diuretic, *n* (%)	13 (54)	1 (5)	<0.001
Platelet inhibitor, *n* (%)	8 (33)	19 (100)	<0.001
Statin, *n* (%)	10 (42)	19 (100)	<0.001

hs cTnT, high-sensitive cardiac troponin T; NT-proBNP, N-terminal prohormone of brain natriuretic peptide; LVH, left ventricular hypertrophy; MI, myocardial infarction; RAS, renin–angiotensin system; NYHA, New York Heart Association; ns, not significant.

**Table 2 diagnostics-13-02943-t002:** Cardiac magnetic resonance imaging profile in patients with positive troponin and unobstructed coronary arteries compared to hypertensive urgency.

Variable	Troponin-Negative Controls (*n =* 24)	Troponin-Positive, Unobstructed Coronary Arteries (*n =* 19)	*p*
Indexed LV EDV (mL/m^2^)	65 (59 to 71)	85 (68 to 100)	<0.001
Indexed LV ESV (mL/m^2^)	22 (16 to 32)	36 (24 to 70)	0.005
LV ejection fraction (%)	63 ± 12	52 ± 16	0.019
Indexed LV mass (g/m^2^)	80 (67 to 103)	100 (79 to 133)	0.017
LV systolic dysfunction, *n* (%)	4(17)	10(53)	0.012
Maximum wall thickness (mm)	14 ± 2.4	14.5 ± 2	0.483
LVH, *n* (%)	13 (54)	14 (74)	0.221
Asymmetric LVH, *n* (%)	4 (17)	2 (11)	0.678
Indexed LA volume (mL/m^2^)	30 (28 to 36)	44 (32 to 56)	<0.001
Global T2-w SI ratio	1.4 ± 0.2	1.4 ± 0.1	0.551
Global native T1 (ms)	1026 ± 39	1042 ± 40	0.198
Global T2 relaxation time (ms)	47 ± 2	49 ± 2	0.004
Global ECV (%)	23 ± 3	25 ± 4	0.047
LGE present, *n* (%)	12/24 (50)	14/17 (82)	0.050
Infarct pattern, *n* (%)	0 (0)	5/17 (29)	-
Non-ischaemic, *n* (%)	12/24 (50)	9/17 (53)	0.445

*p* value for test of two proportions. hs cTnT, high-sensitive cardiac troponin T; NT-proBNP, N-terminal prohormone of brain natriuretic peptide; LV, left ventricular; LV EDV, left ventricular end-diastolic volume; LV ESV, left ventricular end-systolic volume; LVH, left ventricular hypertrophy; SI, signal intensity; T2-w, T2-weighted; ECV, extracellular volume; LGE, late gadolinium enhancement.

**Table 3 diagnostics-13-02943-t003:** Spearman’s correlation of cardiac biomarkers in hypertensive crisis with raised troponin and unobstructed coronary arteries.

	Native T1 Time, ms	T2 Relaxation Time, ms	ECV, %
Native T1 time (ms)	-	0.623, *p* = 0.004	ns
T2 relaxation time (ms)	0.623, *p* = 0.004	-	0.645, *p* = 0.044
Indexed LV mass (g/m^2^)	ns	ns	ns
NT-proBNP (ng/L)	ns	ns	ns

ECV, extracellular volume; NT-proBNP, N-terminal prohormone of brain natriuretic peptide; ns, not significant.

**Table 4 diagnostics-13-02943-t004:** Factors associated with troponin-positive and unobstructed coronary artery.

	Univariate		Multivariate	
	OR (95% CI)	*p*	OR (95% CI)	*p*
Age (years)	1.064 (1.008 to 1.122)	0.024	ns	ns
Creatinine (µmol/L)	1.034 (1.004 to 1.064)	0.024	1.041 (0.991 to 1.093)	0.111
LDH (U/L)	1.013 (1.002 to 1.024)	0.025	ns	ns
NT-proBNP (ng/L)	1.001 (1.000 to 1.002)	0.117	-	-
Indexed LV EDV (mL/m^2^)	1.067 (1.014 to 1.123)	0.012	1.089 (1.010 to 1.173)	0.026
LV ejection fraction (%)	0.943 (0.896 to 0.992)	0.024	ns	ns
Indexed LV mass (g/m^2^)	1.024 (1.001 to 1.048)	0.024	0.959 (0.914 to 1.005)	0.080
LVH, *n* (%)	2.369 (0.646 to 8.685)	0.193	-	-
LGE, *n* (%)	4.667 (1.061 to 20.533)	0.042	2.739 (0.291 to 25.748)	0.378
Native T1 (ms)	1.011 (0.994 to 1.027)	0.198	-	-
T2 relaxation time (ms)	1.925 (1.160 to 3.193)	0.011	2.097 (1.022 to 4.302)	0.043
T2-w SI ratio	3.334 (0.020 to 158.2)	0.541	-	-
ECV (%)	1.271 (0.988 to 1.634)	0.062	-	-

LDH, lactate dehydrogenase; NT-proBNP, N-terminal prohormone of brain natriuretic peptide; LV, left ventricular; LV EDV, left ventricular end-diastolic volume; LAV, left atrial volume; ECV, extracellular volume; LVH, left ventricular hypertrophy; LGE, late gadolinium enhancement; ns, not significant.

**Table 5 diagnostics-13-02943-t005:** Predictors of troponin-positive unobstructed coronary arteries.

	AUC (95% CI)	Sensitivity	Specificity	+LR	−LR	Criteria	*p*
Age (years)	0.71 (0.549 to 0.836)	95	42	1.62	0.13	>43	0.0090
Creatinine (µmol/L)	0.74 (0.578 to 0.857)	90	63	2.39	0.17	>85	0.0035
NT-proBNP (ng/L)	0.71 (0.544 to 0.838)	50	87	3.83	0.58	>477	0.0128
Indexed LV EDV (mL/m^2^)	0.80 (0.653 to 0.908)	95	54	2.07	0.10	>64.9	<0.0001
Lactate dehydrogenase (U/L)	0.72 (0.550 to 0.852)	93	44	1.65	0.15	>207	0.0091
Indexed LAV (mL/m^2^)	0.82 (0.671 to 0.919)	95	50	1.89	0.11	>29.7	<0.0001
Indexed LV mass (g/m^2^)	0.71 (0.555 to 0.840)	68	70	2.35	0.45	>92.1	0.0073
LV ejection fraction (%)	0.70 (0.538 to 0.828)	53	75	2.11	0.63	55	0.0183
Global native T1 time (ms)	0.61 (0.449 to 0.755)	52	83	3.16	0.57	>1048	0.2269
Global T2 relaxation time (ms)	0.71 (0.552 to 0.838)	84	58	2.02	0.27	>47.2	0.0086
Mid-ventricular T2 relaxation time (ms)	0.73 (0576 to 0856)	84	46	1.55	0.34	>47.7	0.0031
Global ECV fraction * (%)	0.71 (0.518 to 0.852)	70	73	2.57	0.41	>23	0.0602
Mid-ventricular ECV (%)	0.77 (0.586 to 0.898)	70	77	3.08	0.39	>23	0.0061

AUC, area under the curve; ECV, extracellular volume; EDV, end-diastolic volume; LAV, left atrial volume; LV, left ventricular; LV EDV, left ventricular end-diastolic volume; NT-proBNP, N-terminal prohormone of brain natriuretic peptide. * ECV available for 32 participants.

## Data Availability

The authors declare their willingness to share the data for this research upon a reasonable request and approval by Stellenbosch University.

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
