# Peer review of "Myocardial Tissue Characterization in Patients with Hypertensive Crisis, Positive Troponin, and Unobstructed Coronary Arteries: A Cardiovascular Magnetic Resonance-Based Study"

_diagnostics, 2023, doi:10.3390/diagnostics13182943_

Round 1

Reviewer 1 Report

Mohammed A Talle et al. used CMR to study 19 patients with elevated troponin and unobstructed coronary arteries and 24 controls. Troponin-positive patients had longer T2 times and higher ECV, indicating myocardial injury with edema. LGE was more common in the positive group. Left ventricular and left atrial volumes were strong predictors of troponin positivity. About one-third of the positive group had CMR evidence of myocardial infarction, suggesting an additional mechanism of injury unrelated to coronary disease. 

Kudos to the Authors for exploring this understudied topic. The use of CMR in hypertensive crises with non-obstructive coronary arteries could be highly valuable and beneficial in clinical practice. Their work sheds light on the potential benefits of this approach, which could significantly improve patient care in critical situations.

The main limitations of the study include the small sample size, the absence of coronary anatomy evaluation in the control group, and the lack of a clear distinction between acute and chronic LGE.

Major issues: 

Abstract 

line 22: improve the terms. E.g T2 mapping values or T2 relaxation times

Line 34-35. I suggest changing this part of the conclusion. In fact, I suggest to underscore that the role of CMR in this setting could be clinically useful. 

Introduction

Line 71. Currently, the evidence of CMR in MINOCA is updated. In fact, takotsubo syndrome is not a common finding in this setting. In addition, early CMR has a proven prognostic role in ischemic MINOCA. (DOI https://doi.org/10.1016/j.jcmg.2023.05.016).

Methods

-       Was the radiologist/cardiologist blinded to clinical data before the CMR report/analyses? If not, this should be added in the limitations section and questions the study design and reliability.

-       Line 159. I believe that also the presence of edema should be considered in the diagnosis of acute myocardial infarction, not only the presence of MVO + ischemic LGE.

-       The study design has some biases. 1) Patients with negative troponin did not undergo coronary angiography to exclude those with possible obstructive CAD. 2) The exclusion criteria did not include previous AMI. The presence of LGE at the CMR without concomitant edema may represent a chronic scar rather than an acute AMI. In fact, the prevalence of diabetes was greater in the controls group and 10/60 patients had a class NNYHA III-IV maybe representing a chronic HF population. 

Results

-       Table 3. Why native T1 is not correlated with ECV?  How do you explain this?

-       Could be useful to describe the CMR findings in patients with positive delta troponin (15 of the total).

-      Table 2. From the reader's point of view, it might be useful to simplify the table by including some data in the supplements. 

-       Table 4, in the logistic binary regression an overfitting bias is possible. In fact, compared to the overall populations too many covariates are entered in the model. This should be acknowledged in the limitations section.

Discussion

-       I suggest to tone-down the subsequent sentence: “5) Indexed LVEDV and LAV demonstrated 316 a strong diagnostic utility for predicting troponin-positive unobstructed coronary arteries 317 using the ROC curve, followed by ECV and T2 times.” T2 values emerged as the primary predictor of troponin-positive hypertensive crisis, indicating a potential link to myocardial injury in this setting. This crucial result warrants in-depth discussion in the manuscript, shedding light on its biological and imaging implications and offering avenues for future research and clinical applications.

-       I really appreciate the practical messages in the discussion section. Please note that the evidence on the use of CMR in MINOCA has increased. Also, consider that second-level echocardiography including strain could be useful in patients with hypertensive crisis to reveal subclinical CAD (e.g. 10.1093/ehjci/jead046). Note that also coronary CT (including FFR and perfusion) could be useful and really informative in this setting

-       Finally, also coronary microcirculation could be impaired in patients with hypertension and could determine the acute myocardial injury.

Limitations

- The sample size is very limited. Please add this aspect in the limitations section.

Conclusion

- I suggest making it more explicit that the use of CMR in this context could be useful.

Additional suggestions: 

- Figure 2. Please add the colour scale also for the T1 mapping boxes. 

- Line 218. Please, correct the double space.

Minor English revision is required. I suggest double-checking abbreviations.

Author Response

Response to reviewer comments_1

Reviewer comment 1: Kudos to the Authors for exploring this understudied topic. The use of CMR in hypertensive crises with non-obstructive coronary arteries could be highly valuable and beneficial in clinical practice. Their work sheds light on the potential benefits of this approach, which could significantly improve patient care in critical situations.

Author response 1: Thank you for these positive comments.

Reviewer comment 2: The main limitations of the study include the small sample size, the absence of coronary anatomy evaluation in the control group, and the lack of a clear distinction between acute and chronic LGE.

Author response 2: Thank you for this observation. We have included the small sample size and lack of coronary anatomy evaluation in the limitations, as suggested. Reviewer comment 3: Sex-specific differences should be mentioned/discussed in the limitations. (Lines 723-747). 

Reviewer comment 3: line 22: improve the terms. E.g T2 mapping values or T2 relaxation times.

Author response 3: We have uniformly adopted T2 relaxation time. (Lines 22-31, and across the manuscript).

Reviewer comment 4: Line 34-35. I suggest changing this part of the conclusion. In fact, I suggest to underscore that the role of CMR in this setting could be clinically useful.

Author response 4: Thank you for the comment. We have revised the conclusion, underscoring the clinical utility of CMR as suggested.( Lines 33-35).

Reviewer comment 5: Line 71. Currently, the evidence of CMR in MINOCA is updated. In fact, takotsubo syndrome is not a common finding in this setting. In addition, early CMR has a proven prognostic role in ischemic MINOCA. (DOI https://doi.org/10.1016/j.jcmg.2023.05.016).

Author response 5: Thank you for drawing our attention to this important concept. We have reflected the recent evidence for the beneficial role of early CMR in MINOCA. We agree with the observation regarding Takotsubo. Nonetheless, the reference is maintained in a bid to bring aetiologic diagnoses established in the setting of raised troponin and unobstructed coronary artery (Lines 94-96).

Reviewer comment 6: Was the radiologist/cardiologist blinded to clinical data before the CMR report/analyses? If not, this should be added in the limitations section and questions the study design and reliability.

Author response 6: Thank you. We have added this as a limitation. Although the cardiologist was not blinded to recruitment of participants, steps were taken at the a priori, to minimise reporting bias. Enrolment into the study lasted two years, with an interval of six months from the last participant to the time the imaging data was analysed. The images were anonymised to minimise further reporting bias, and post-processing was carried out for the group as a single cluster without being classified as cases and controls (Lines 192-195; Lines 746-747).

Reviewer comment 7: Line 159. I believe that also the presence of edema should be considered in the diagnosis of acute myocardial infarction, not only the presence of MVO + ischemic LGE.

Author response 7: We agree with the reviewer on this. Oedema was indeed considered in the diagnosis of myocardial infarction (Line 268).

Reviewer comment 8: The study design has some biases. 1) Patients with negative troponin did not undergo coronary angiography to exclude those with possible obstructive CAD. 2) The exclusion criteria did not include previous AMI. The presence of LGE at the CMR without concomitant edema may represent a chronic scar rather than an acute AMI. In fact, the prevalence of diabetes was greater in the controls group and 10/60 patients had a class NNYHA III-IV maybe representing a chronic HF population. 

Author response 8: Thank you raising the discussion about potential bias in this study. We have addressed this in the limitations. Although AMI was not an exclusion criterion, there was only one case of previous MI in the cohort. We agree with the reviewer’s observation regarding the NYHA. However, this was mainly driven by pulmonary edema associated with acute severe hypertension and responded promptly to the management of hypertensive emergency. We also realise that acute severe hypertension can precipitate acute heart failure presentation in individuals with chronic compensated heart failure (Lines 723 – 747).

Reviewer comment 9: Table 3. Why native T1 is not correlated with ECV?  How do you explain this?

Author response 9: Hypertensive emergency is associated with both intracellular and interstitial oedema. We therefore considered that the lack of correlation may be related to the differential involvement of the intracellular and interstitial compartments in terms of myocardial edema. This may result in a variable quantification of myocardial water content by native T1 time and ECV since the latter is limited to the extracellular interstitial compartment (Lines 565-567).

Reviewer comment 10: Could be useful to describe the CMR findings in patients with positive delta troponin (15 of the total).

Author response 10: Thank you for this suggestion. We have provided a comparison of CMR findings in the delta 20% group and the controls in Table S3. However, the small sample size precluded a subgroup comparison between those with and without delta 20% (Lines 405-410 and Table S3 (Supplementary material)).

Reviewer comment 11: Table 2. From the reader's point of view, it might be useful to simplify the table by including some data in the supplements. 

Author response 11: We have modified Table 2 by including only the global parametric mapping variables. The remaining variables are included in Table S1 (Lines 411-416; Table S1 (Supplementary material)).

Reviewer comment 12: Table 4, in the logistic binary regression an overfitting bias is possible. In fact, compared to the overall populations too many covariates are entered in the model. This should be acknowledged in the limitations section.

Author response 12: Thank you. We have included this as a limitation (Lines 726-727).

Reviewer comment 13: I suggest to tone-down the subsequent sentence: “5) Indexed LVEDV and LAV demonstrated 316 a strong diagnostic utility for predicting troponin-positive unobstructed coronary arteries 317 using the ROC curve, followed by ECV and T2 times.” T2 values emerged as the primary predictor of troponin-positive hypertensive crisis, indicating a potential link to myocardial injury in this setting. This crucial result warrants in-depth discussion in the manuscript, shedding light on its biological and imaging implications and offering avenues for future research and clinical applications.

Author response 13: We have revised the sentence to read: Furthermore, indexed LVEDV and LAV demonstrated a diagnostic utility for predicting troponin-positive unobstructed coronary arteries using the ROC curve, followed by ECV and T2 times. We have also added a perspective linking T2 to myocardial injury and myocardial oedema, as suggested (Lines 528-530; Lines 571-631).

Reviewer comment 14: I really appreciate the practical messages in the discussion section. Please note that the evidence on the use of CMR in MINOCA has increased. Also, consider that second-level echocardiography including strain could be useful in patients with hypertensive crisis to reveal subclinical CAD (e.g. 10.1093/ehjci/jead046). Note that also coronary CT (including FFR and perfusion) could be useful and really informative in this setting.

Author response 14: Thank you for these generous and positive comments. We have included the evidence for the prognostic role of early CMR in patients with MINOCA in the discussion. We have included not having FFR, IVUS, OCT, and provocative test as limitations (Lines 717-722; Lines 730-739).

Reviewer comment 15: Finally, also coronary microcirculation could be impaired in patients with hypertension and could determine the acute myocardial injury.

Author response 15: Thank you for this observation. This has been highlighted in the discussion. We also included not doing myocardial perfusion study as a limitation (Line 545; Line 737-739).

Reviewer comment 16: The sample size is very limited. Please add this aspect in the limitations section.

Author response 16: We have included the small sample size as a limitation (Line 724).

Reviewer comment 17: I suggest making it more explicit that the use of CMR in this context could be useful.

Author response 17: Thank you for the suggestion. We have revised the conclusion, accordingly, underscoring the potential utility of CMR in the evaluation and management of patients with hypertensive crisis (Lines 776-781).

Reviewer comment 18: Figure 2. Please add the colour scale also for the T1 mapping boxes. 

Author response 18: We have updated the Figure by adding the scale for T1 mapping (Line 429).

Reviewer comment 19: Line 218. Please, correct the double space.

Author response 19: Corrected (Line 375).

Reviewer comment 20: Minor English revision is required. I suggest double-checking abbreviations.

Author response 20: The manuscript has been revised, abbreviations checked, and all the changes made have been highlighted with track changes.

Reviewer 2 Report

The present work by Talle et al., provides an analysis of characterization and correlation of patients with hypertensive crisis,elevated cardiac troponin and unobstructed cardiac arteries with cardiac tissue biomakers and cardiac magnetic resonance (CMR)-functional parameters. Their major results include identifying a positive correlation in troponin-positive hypertensive crisis patient with CMR markers (increased T2 time and extracellular cardiac volume); suggestive of myocardial infarction with oedema.  Importantly, a sub-group of these patients had markers that were more indicative of myocardial fibrosis/necrosis which suggests additonal mechanisms of myocardial injury that contributes to the pathological phenotype of hypertensive crisis without obstructive coronary disease.

The paper follows a well structured pattern of study which effectively aims to answer the main hypothesis. Methods are well described and statistical analysis employed are appropiately described and used. While the study is mostly focused on correlations and associative descriptions (without strong emphasis on the mechanistic insight), these are of importance due to the lack of CMR-based characterization of this particular subset of patients with hypertensive crisis that also present elevated troponin levels with unobstructed coronary arteries. Thus, this reviewer finds this work to be of potential interest.

My only comments are stated below:

1) The differences in between the two different sub-sets of troponin + patients based on their late gadolinium enhancement (infarct pattern LGE vs nonischaemic LGE) is an interesting point of study that could benefit from further investigation. Could the authors analyze these two subsets of patients and compare their CMR profile and serum biomarkers? Could this provide additional information on the potential mechanism involved in the variability of clinical signs in patients with hypertensive crisis and troponin positive levels?

2) Sex-specific differences should be mentioned/discussed in the limitations. 

3) Line 188: The troponin-positive group with unobstructed coronary arteries were on average 10  years younger than the troponin-negative group. Do you mean that they were 10 years older?

Author Response

Response to reviewers comments

Reviewer 2

Reviewer comment 1: The paper follows a well-structured pattern of study which effectively aims to answer the main hypothesis. Methods are well described and statistical analysis employed are appropriately described and used. While the study is mostly focused on correlations and associative descriptions (without strong emphasis on the mechanistic insight), these are of importance due to the lack of CMR-based characterization of this particular subset of patients with hypertensive crisis that also present elevated troponin levels with unobstructed coronary arteries. Thus, this reviewer finds this work to be of potential interest.

Author response 1: Thank you for these positive comments.

Reviewer comment 2: The differences in between the two different sub-sets of troponin + patients based on their late gadolinium enhancement (infarct pattern LGE vs nonischaemic LGE) is an interesting point of study that could benefit from further investigation. Could the authors analyse these two subsets of patients and compare their CMR profile and serum biomarkers? Could this provide additional information on the potential mechanism involved in the variability of clinical signs in patients with hypertensive crisis and troponin positive levels?

Author response 2: Thank you for the suggestion. We have compared CMR and biomarkers in the two categories of LGE. However, the analysis is prone to type 2 statistical error due to the small sample size. Nonetheless, we have included this analysis in the results, and the discussion (Lines 447-460 and Table S4.; Lines 669-671).

Reviewer comment 3: Sex-specific differences should be mentioned/discussed in the limitations.

Author response 3: We have updated the limitations to reflect this (Line 724-725).

Reviewer comment 4: Line 188: The troponin-positive group with unobstructed coronary arteries were on average 10  years younger than the troponin-negative group. Do you mean that they were 10 years older?

Author response 4: Thank you, this error has now been corrected. We have changed younger to older (Line 298).

Round 2

Reviewer 1 Report

The authors have responded to my questions and have overall improved the manuscript.

Minor English revision is recommended.